# MAKE LEAD BIAS IN YOUR FAVOR: A SIMPLE AND EFFECTIVE METHOD FOR NEWS SUMMARIZATION

## ABSTRACT

Lead bias is a common phenomenon in news summarization, where early parts of an article often contain the most salient information. While many algorithms exploit this fact in summary generation, it has a detrimental effect on teaching the model to discriminate and extract important information. We propose that the lead bias can be leveraged in a simple and effective way in our favor to pretrain abstractive news summarization models on large-scale unlabeled corpus: predicting the leading sentences using the rest of an article. Via careful data cleaning and filtering, our transformer-based pretrained model without any finetuning achieves remarkable results over various news summarization tasks. With further finetuning, our model outperforms many competitive baseline models. For example, the pretrained model without finetuning achieves state-of-the-art results on DUC-2003 and DUC-2004 datasets. The finetuned model obtains 3.2% higher ROUGE-1, 1.6% higher ROUGE-2 and 2.1% higher ROUGE-L scores than the best baseline model on XSum dataset. Human evaluations further show the effectiveness of our method.

## 1    INTRODUCTION

The goal of text summarization is to condense a piece of text into a shorter version that contains the salient information. Due to the prevalence of news articles and the need to provide succinct summaries for readers, a majority of existing datasets for summarization come from the news domain (Hermann et al., 2015; Sandhaus, 2008; Narayan et al., 2018). However, according to journalistic conventions, the most important information in a news report usually appears near the beginning of the article (Kedzie et al., 2018; Jung et al., 2019). While it facilitates faster and easier understanding of the news for readers, this lead bias causes undesirable consequences for summarization models. The output of these models is inevitably affected by the positional information of sentences. Furthermore, the simple baseline of using the top few sentences as summary can achieve a stronger performance than many sophisticated models (See et al., 2017). It can take a lot of effort for models to overcome the lead bias Kedzie et al. (2018).

Additionally, most existing summarization models are fully supervised and require time and labor-intensive annotations to feed their insatiable appetite for labeled data. For example, the New York Times Annotated Corpus (Sandhaus, 2008) contains 1.8 million news articles, with 650,000 summaries written by library scientists. Therefore, some recent work (Gusev, 2019) explores the effect of domain transfer to utilize datasets other than the target one. But this method may be affected by the domain drift problem and still suffers from the lack of labelled data.

The recent promising trend of pretraining models (Devlin et al., 2018; Radford et al., 2018) proves that a large quantity of data can be used to boost NLP models' performance. Therefore, we put forward a novel method to leverage the lead bias of news articles in our favor to conduct large-scale pretraining of summarization models. The idea is to leverage the top few sentences of a news article as the target summary and use the rest as the content. The goal of our pretrained model is to generate an abstractive summary given the content. Coupled with careful data filtering and cleaning, the lead bias can provide a delegate summary of sufficiently good quality, and it immediately renders the large quantity of unlabeled news articles corpus available for training news summarization models.

We employ this pretraining idea on a three-year collection of online news articles. We conduct thorough data cleaning and filtering. For example, to maintain a quality assurance bar for using

leading sentences as the summary, we compute the ratio of overlapping non-stopping words between the top 3 sentences and the rest of the article. As a higher ratio implies a closer semantic connection, we only keep articles for which this ratio is higher than a threshold.

We end up with 21.4M articles based on which we pretrain a transformer-based encoder-decoder summarization model. We conduct thorough evaluation of our models on five benchmark news summarization datasets. Our pretrained model achieves a remarkable performance on various target datasets without *any* finetuning. This shows the effectiveness of leveraging the lead bias to pretrain on large-scale news data. We further finetune the model on target datasets and achieve better results than a number of strong baseline models. For example, the pretrained model without finetuning obtains state-of-the-art results on DUC-2003 and DUC-2004. The finetuned model obtains 3.2% higher ROUGE-1, 1.6% higher ROUGE-2 and 2.1% higher ROUGE-L scores than the best baseline model on XSum dataset (Narayan et al., 2018). Human evaluation results also show that our models outperform existing baselines like pointer-generator network.

The rest of paper is organized as follows. We introduce related work in news summarization and pretraining in Section 2. We describe the details of pretraining using lead bias in Section 3. We introduce the transformer-based summarization model in Section 4. We show the experimental results in Section 5 and conclude the paper in Section 6.

## 2 RELATED WORK

### 2.1 DOCUMENT SUMMARIZATION

End-to-end abstractive text summarization has been intensively studied in recent literature. To generate summary tokens, most architectures take the encoder-decoder approach (Sutskever et al., 2014). Rush et al. (2015b) first introduces an attention-based seq2seq model to the abstractive sentence summarization task. However, its output summary degenerates as document length increases, and out-of-vocabulary (OOV) words cannot be efficiently handled. To tackle these challenges, See et al. (2017) proposes a pointer-generator network that can both produce words from the vocabulary via a generator and copy words from the source article via a pointer. Paulus et al. (2017); Li et al. (2018) utilize reinforcement learning to improve the result. Gehrmann et al. (2018) uses a content selector to over-determine phrases in source documents that helps constrain the model to likely phrases. You et al. (2019) adds Gaussian focal bias and a salience-selection network to the transformer encoder-decoder structure (Vaswani et al., 2017) for abstractive summarization. Grenander et al. (2019) randomly reshuffles the sentences in news articles to reduce the effect of lead bias in extractive summarization.

### 2.2 PRETRAINING

In recent years, pretraining language models have proved to be quite helpful in NLP tasks. The state-of-the-art pretrained models include CoVe (McCann et al., 2017), ELMo (Peters et al., 2018), GPT (Radford et al., 2018), BERT (Devlin et al., 2018) and UniLM (Dong et al., 2019). Built upon large-scale corpora, these pretrained models learn effective representations for various semantic structures and linguistic relationships. As a result, pretrained models have been widely used with considerable success in applications such as question answering (Zhu et al., 2018), sentiment analysis (Peters et al., 2018) and passage reranking (Nogueira & Cho, 2019). Furthermore, UniLM (Dong et al., 2019) leverages its sequence-to-sequence capability for abstractive summarization; the BERT model has been employed as an encoder in BERTSUM (Liu & Lapata, 2019) for extractive/abstractive summarization.

Compared to our work, UniLM (Dong et al., 2019) is a general language model framework and does not take advantage of the special semantic structure of news articles. Similarly, BERTSUM (Liu & Lapata, 2019) directly copies the pretrained BERT structure into its encoder and finetunes on labelled data instead of pretraining with the large quantity of unlabeled news corpus available.

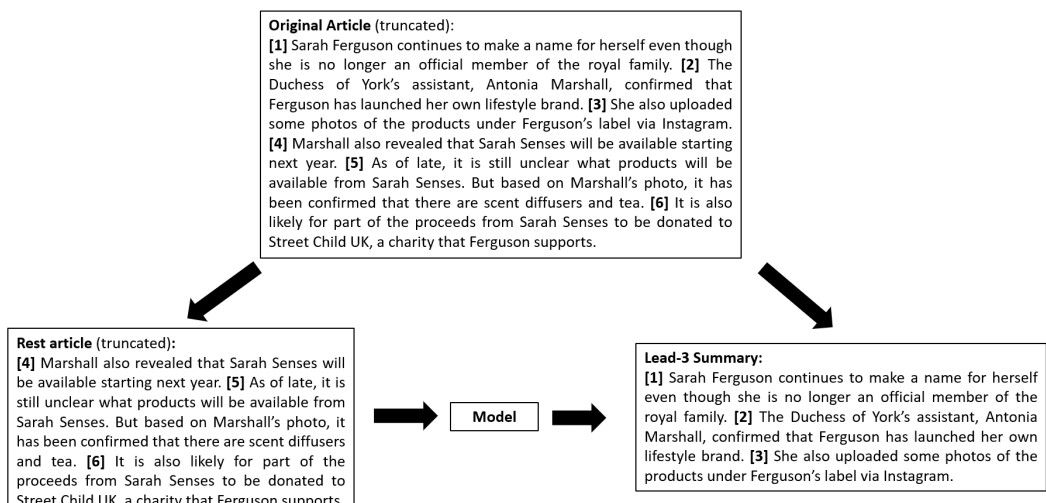

Figure 1: Using Lead-3 summary as target in pretraining.

# 3 Pretraining with Leading Sentences

News articles usually follow the convention of placing the most important information early in the content, forming an inverted pyramid structure. This lead bias has been discovered in a number of studies (Kedzie et al., 2018; Jung et al., 2019; Grenander et al., 2019). One of the consequences is that the lead baseline, which simply takes the top few sentences as the summary, can achieve a rather strong performance in news summarization. For instance, in the CNN/Daily Mail dataset (Hermann et al., 2015), using the top three sentences as summaries can get a higher ROUGE score than many deep learning based models. This positional bias brings lots of difficulty for models to extract salient information from the article and generate high-quality summaries. For instance, Grenander et al. (2019) discovers that most models' performances drop significantly when a random sentence is inserted in the leading position, or when the sentences in a news article are shuffled.

On the other hand, news summarization, just like many other supervised learning tasks, suffers from the scarcity of labelled training data. Abstractive summarization is especially data-hungry since the efficacy of models depends on high-quality handcrafted summaries.

We propose that the lead bias in news articles can be leveraged in our favor to train an abstractive summarization model without human labels. Given a news article, we treat the top three sentences, denoted by Lead-3, as the target summary, and use the rest of the article as news content. The goal of the summarization model is to produce Lead-3 using the following content, as illustrated in Figure 1.

The benefit of this approach is that the model can leverage the large number of unlabeled news articles for pretraining. In the experiment, we find that the pretrained model alone can have a strong performance on various news summarization datasets, without any further training. We also finetune the pretrained model on downstream datasets with labelled summaries. The model can quickly adapt to the target domain and further increase its performance.

It is worth noting that this idea of utilizing structural bias for large-scale summarization pretraining is not limited to specific types of models, and it can be applied to other types of text as well: academic papers with abstracts, novels with editor's notes, books with tables of contents.

However, one should carefully examine and clean the source data to take advantage of lead bias, as the top three sentences may not always form a good summary. We provide more details in the experiments about the data filtering and cleaning mechanism we apply.

## 4 MODEL

In this section, we introduce our abstractive summarization model, which has a transformer-based encoder-decoder structure. We first formulate the supervised summarization problem and then present the network architecture.

### 4.1 PROBLEM FORMULATION

We formalize the problem of supervised abstractive summarization as follows. The input consists of $a$ pairs of articles and summaries: $\{(X_1, Y_1), (X_2, Y_2), ..., (X_a, Y_a)\}$. Each article and summary are tokenized: $X_i = (x_1, ..., x_{L_i})$ and $Y_i = (y_1, ..., y_{N_i})$. In abstractive summarization, the summary tokens need not be from the article. For simplicity, we will drop the data index subscript. The goal of the system is to generate summary $Y = (y_1, ..., y_m)$ given the transcript $X = \{x_1, ..., x_n\}$.

### 4.2 NETWORK STRUCTURE

We utilize a transformer-based encoder-decoder structure that maximizes the conditional probability of the summary: $P(Y|X, \theta)$, where $\theta$ represents the parameters.

#### 4.2.1 ENCODER

The encoder maps each token into a fixed-length vector using a trainable dictionary $\mathcal{D}$ randomly initialized using a normal distribution with zero mean and a standard deviation of 0.02. Each transformer block conducts multi-head self-attention. And we use sinusoidal positional embedding in order to process arbitrarily long input. In the end, the output of the encoder is a set of contextualized vectors:

$$\text{Encoder-Transformer}(\{x_1, ..., x_n\}) = \{u_1^E, ..., u_n^E\}$$

#### 4.2.2 DECODER

The decoder is a transformer that generates the summary tokens one at a time, based on the input and previously generated summary tokens. Each token is projected onto a vector using the same dictionary $\mathcal{D}$ as the encoder.

The decoder transformer block includes an additional cross-attention layer to fuse in information from the encoder. The output of the decoder transformer is denoted as:

$$\text{Decoder-Transformer}(\{w_1, ..., w_{k-1}\}) = \{u_1^D, ..., u_{k-1}^D\} \tag{1}$$

To predict the next token $w_k$, we reuse the weights of dictionary $\mathcal{D}$ as the final linear layer to decode $u_{k-1}^D$ into a probability distribution over the vocabulary: $P(w_k|w_{<k}, u_{1:m}^E) = \text{softmax}(\mathcal{D}u_{k-1}^D)$.

**Training**. During training, we seek to minimize the cross-entropy loss:

$$L(\theta) = -\frac{1}{m} \sum_{k=1}^{m} \log P(y_k|y_{<k}, X) \tag{2}$$

We use teacher-forcing in decoder training, i.e. the decoder takes ground-truth summary tokens as input. The model has 10 layers of 8-headed transformer blocks in both its encoder and decoder, with 154.4M parameters.

**Inference**. During inference, we employ beam search to select the best candidate. The search starts with the special token ⟨BEGIN⟩. We ignore any candidate word which results in duplicate trigrams. We select the summary with the highest average log-likelihood per token.

## 5 EXPERIMENTS

### 5.1 DATASETS

We evaluate our model on five benchmark summarization datasets: the New York Times Annotated Corpus (NYT) (Sandhaus, 2008), XSum (Narayan et al., 2018), the CNN/DailyMail dataset (Her-

mann et al., 2015), DUC-2003 and DUC-2004 (Over et al., 2007). These datasets contain 104K, 227K, 312K, 624 and 500 news articles and human-edited summaries respectively, covering different topics and various summarization styles. For NYT dataset, we use the same train/val/test split and filtering methods following Durrett et al. (2016). As DUC-2003/2004 datasets are very small, we follow West et al. (2019) to employ them as test set only.

## 5.2 Implementation Details

We use SentencePiece (Kudo & Richardson, 2018) for tokenization, which segments any sentence into subwords. We train the SentencePiece model on pretrained data to generate a vocabulary of size 32K and of dimension 720. The vocabulary stays fixed during pretraining and finetuning.

**Pretraining.** We collect three years of online news articles from June 2016 to June 2019. We filter out articles overlapping with the evaluation data on media domain and time range. We then conduct several data cleaning strategies.

First, many news articles begin with reporter names, media agencies, dates or other contents irrelevant to the content, e.g. "New York (CNN) –", "Jones Smith, May 10th, 2018:". We therefore apply simple regular expressions to remove these prefixes.

Second, to ensure that the summary is concise and the article contains enough salient information, we only keep articles with 10-150 words in the top three sentences and 150-1200 words in the rest, and that contain at least 6 sentences in total. In this way, we filter out i) articles with excessively long content to reduce memory consumption; ii) very short leading sentences with little information which are unlikely to be a good summary. To encourage the model to generate abstractive summaries, we also remove articles where any of the top three sentences is exactly repeated in the rest of the article.

Third, we try to remove articles whose top three sentences may not form a relevant summary. For this purpose, we utilize a simple metric: overlapping words. We compute the portion of non-stopping words in the top three sentences that are also in the rest of an article. A higher portion implies that the summary is representative and has a higher chance of being inferred by the model using the rest of the article. To verify, we compute the overlapping ratio of non-stopping words between human-edited summary and the article in CNN/DailyMail dataset, which has a median value of 0.87. Therefore, in pretraining, we keep articles with an overlapping word ratio higher than 0.65.

These filters rule out around 95% of the raw data and we end up with 21.4M news articles, 12,000 of which are randomly sampled for validation.

We pretrain the model for 10 epochs and evaluate its performance on the validation set at the end of each epoch. The model with the highest ROUGE-L score is selected.

During pretraining, we use a dropout rate of 0.3 for all inputs to transformer layers. The batch size is 1,920. We use RAdam (Liu et al., 2019) as the optimizer, with a learning rate of $10^{-4}$. Also, due to the different numerical scales of the positional embedding and initialized sentence piece embeddings, we divide the positional embedding by 100 before feeding it into the transformer. The beam width is set to 5 during inference.

**Finetuning.** During finetuning, we keep the optimizer, learning rate and dropout rate unchanged as in pretraining. The batch size is 32 for all datasets. We pick the model with the highest ROUGE-L score on the validation set and report its performance on the test set. More details are given in the Appendix.

Our strategy of Pretraining with unlabeled Lead-3 summaries is called **PL**. We denote the pretrained model with finetuning on target datasets as **PL-FT**. The model with only pretraining and no finetuning is denoted as **PL-NoFT**, which is the same model for all datasets.

## 5.3 Baseline

To compare with our model, we select a number of strong summarization models as baseline systems. Lead-X uses the top $X$ sentences as a summary (Liu & Lapata, 2019). The value of $X$ is

| Model | R1 | R2 | RL | Model | R1 | R2 | RL |
|---|---|---|---|---|---|---|---|
| LEAD-3 | 39.58 | 20.11 | 35.78 | LEAD-1 | 16.30 | 1.60 | 11.95 |
| PTGEN | 42.47 | 25.61 | — | PTGEN | 29.70 | 9.21 | 23.24 |
| PTGEN + COV | 43.71 | 26.40 | — | PTGEN+COV | 28.10 | 8.02 | 21.72 |
| DRM | 42.94 | 26.02 | — | TCONVS2S | 31.89 | 11.54 | 25.75 |
| PL-NoFT | 35.32 | 17.80 | 31.88 | PL-NoFT | 24.12 | 5.59 | 19.20 |
| PL-FT | **44.18**$^*$ | **27.49**$^*$ | **40.65**$^{**}$ | PL-FT | **35.06**$^{**}$ | **13.12**$^{**}$ | **27.86**$^{**}$ |

Table 1: ROUGE recall scores on **NYT** test set. Table 2: ROUGE F1 results on **XSum** test set.

| Model | R1 | R2 | RL |
|---|---|---|---|
| LEAD-3 | 40.5 | 17.7 | 36.7 |
| **Unsupervised** | | | |
| SEQ$^3$ | 17.85 | 3.94 | 19.53 |
| GPT-2 | 29.34 | 8.27 | 26.58 |
| PL-NoFT | **38.95**$^{**}$ | **16.27**$^{**}$ | **35.11**$^{**}$ |
| **Supervised** | | | |
| PTGEN | 36.44 | 15.66 | 33.42 |
| PTGEN+COV | 39.53 | 17.28 | 36.38 |
| DRM | 39.87 | 15.82 | 36.90 |
| BOTTOMUP | **41.22** | **18.68** | **38.34** |
| PL-FT | 40.41 | 17.81 | 37.19 |

Table 3: ROUGE F1 results on **CNN/DailyMail** test set.

| | **DUC-2003** | | | **DUC-2004** | | |
|---|---|---|---|---|---|---|
| Model | R1 | R2 | RL | R1 | R2 | RL |
| **Supervised** | | | | | | |
| ABS | 28.48 | 8.91 | 23.97 | 28.18 | 8.49 | 23.81 |
| DRGD | / | / | / | 31.79 | 10.75 | 27.48 |
| **Unsupervised** | | | | | | |
| SEQ$^3$ | 20.90 | 6.08 | 18.55 | 22.13 | 6.18 | 19.3 |
| BottleSum$^{Ex}$ | 21.80 | 5.63 | 19.19 | **22.85** | 5.71 | 19.87 |
| BottleSum$^{Self}$ | 21.54 | 5.93 | 18.96 | 22.30 | 5.84 | 19.60 |
| GPT-2 | 4.98 | 0.37 | 4.63 | 5.29 | 0.28 | 4.90 |
| PL-NoFT | **23.23**$^{**}$ | **6.64**$^{**}$ | **20.42**$^{**}$ | 22.71 | **6.37**$^*$ | **19.97** |

Table 4: ROUGE recall scores on **DUC-2003** and **DUC-2004** test set.

3 for NYT and CNN/DailyMail[1] and 1 for XSum to accommodate the nature of summary length. PTGEN (See et al., 2017) is the pointer-generator network. DRM (Paulus et al., 2017) leverages deep reinforcement learning for summarization. TCONVS2S (Narayan et al., 2018) is based on convolutional neural networks. BOTTOMUP (Gehrmann et al., 2018) uses a bottom-up approach to generate summarization. ABS (Rush et al., 2015a) uses neural attention for summary generation. DRGD (Li et al., 2017) is based on a deep recurrent generative decoder.

To compare with our pretrain-only model, we include several unsupervised abstractive baselines: SEQ$^3$ (Baziotis et al., 2019) employs the reconstruction loss and topic loss for summarization. BottleSum (West et al., 2019) leverages unsupervised extractive and self-supervised abstractive methods. GPT-2 (Radford et al., 2018) is a large-scaled pretrained language model which can be directly used to generate summaries[2].

---

[1]The ROUGE scores here on CNN/Daily Mail are higher than those reported in the original paper, because we extract 3 sentences in Daily Mail rather than 4.

[2]We follow GPT-2's approach to add *TL;DR:* after the article for summary generation. And we use the GPT-2 small model available.

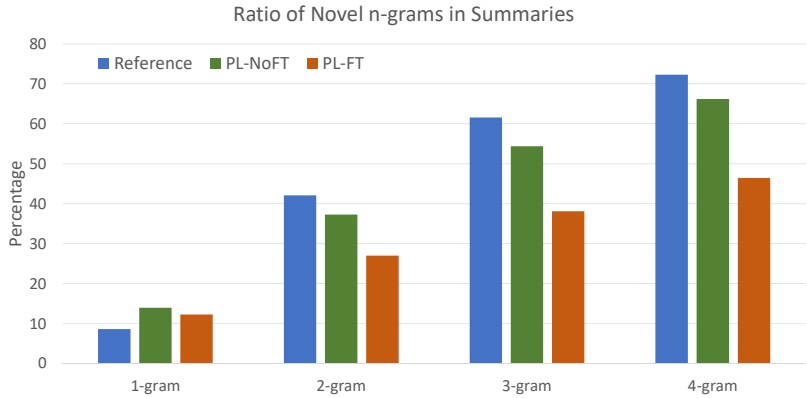

Figure 2: Ratio of novel n-grams in summaries from reference, PL-NoFT and PL-FT models in NYT test set.

## 5.4 METRICS

We employ the standard ROUGE-1, ROUGE-2 and ROUGE-L metrics (Lin, 2004) to evaluate all summarization models. These three metrics respectively evaluate the accuracy on unigrams, bigrams and longest common subsequence. ROUGE metrics have been shown to highly correlate with the human judgment (Lin, 2004). Following Durrett et al. (2016); West et al. (2019), we use F-measure ROUGE on XSUM and CNN/DailyMail, and use limited-length recall-measure ROUGE on NYT and DUC. In NYT, the prediction is truncated to the length of the ground-truth summaries; in DUC, the prediction is truncated to 75 characters.

## 5.5 RESULTS

The results are displayed in Table 1, Table 2, Table 3 and Table 4. As shown, on both NYT and XSum dataset, PL-FT outperforms all baseline models by a large margin. For instance, PL-FT obtains 3.2% higher ROUGE-1, 1.6% higher ROUGE-2 and 2.1% higher ROUGE-L scores than the best baseline model on XSum dataset. We conduct statistical test and found that the results are all significant with p-value smaller than 0.05 (marked by *) or 0.01 (marked by **), compared with previous best scores. On CNN/DailyMail dataset, PL-FT outperforms all baseline models except BottomUp (Gehrmann et al., 2018).

PL-NoFT, the pretrained model without any finetuning, also gets remarkable results. On XSum dataset, PL-NoFT is almost 8% higher than Lead-1 in ROUGE-1 and ROUGE-L. On CNN/DailyMail dataset, PL-NoFT significantly outperforms unsupervised models SEQ$^3$ and GPT-2, and even surpasses the supervised pointer-generator network. PL-NoFT also achieves state-of-the-art results on DUC-2003 and DUC-2004 among unsupervised models (except ROUGE-1 on DUC-2004), outperforming other carefully designed unsupervised summarization models. It's worth noting that PL-NoFT is the same model for all experiments, which proves that our pretrain strategy is effective across different news corpus.

## 5.6 ANALYSIS

**Example Summaries.** We showcase two examples of LEAD-3, reference summaries and the summaries generated by our models in Figure 3. As shown, even without finetuning on the target dataset, PL-NoFT model can generate summaries with salient information, often shortening and reorganizing the original article's contents. After further finetuning, PL-FT captures more relevant content from the article that is similar to that in the reference summary.

**Abstractiveness.** We measure the abstractiveness of our model via the ratio of novel n-grams in summaries, i.e. the percentage of n-grams in the summary that are not present in the article. Figure 2 shows this ratio in summaries from reference and generated by PL-NoFT and PL-FT in NYT dataset.

**Lead-3:**
china's recent test of an antisatellite weapon sent a confusing message to the world about its military intentions, but the united states and china are slowly building stronger military-to-military ties, the top-ranking united states military officer said friday.
gen. peter pace, chairman of the joint chiefs of staff, said he urged his chinese counterparts in meetings on thursday and friday to provide more information about the aims of their military buildup.
he called the meetings "encouraging" but said he did not get concrete answers.

**Reference:**
gen peter pace says he urged his chinese counterparts to provide more information about aims of their military buildup.
calls meetings in beijing encouraging, but says he did not get concrete answers.
says china's recent test of antisatellite weapon sent confusing message to world about its military intentions, but us and china are slowly building stronger military-to-military ties.

**PL-NoFT Summary:**
china's recent test of an antisatellite weapon sent a confusing message to the world about its military intentions, but the united states and china are slowly building stronger military-to-military ties. the test, which china said nothing about for more than a week, raised alarms in washington that the chinese military might seek the capacity to cripple the pentagon.

**PL-FT Summary:**
chairman gen peter pace urges chinese counterparts in meetings to provide more information about aims of their military buildup.
china's recent test of antisatellite weapon sends confusing message to world about its military intentions, but us and china are slowly building stronger military-to-military ties.

**Lead-3:**
in the first major investigation of medicare marketing, the oklahoma insurance commissioner has documented widespread misconduct by agents working for humana and has ordered the company to take corrective action to protect consumers against high-pressure sales tactics.
the commissioner, kim holland, said some agents had enrolled medicare recipients in humana products that "they did not understand and did not want."
at least 68 agents did not have the licenses needed to sell insurance in oklahoma, ms. holland said monday in an interview.

**Reference:**
oklahoma insurance commissioner kim holland, in first major probe of medicare marketing, documents widespread misconduct by agents working for humana.
orders company to take corrective action to protect consumers against high-pressure sales tactics.
contends some agents enrolled medicare recipients in humana products they did not understand or want.

**PL-NoFT Summary:**
the state's top insurance regulator said monday that it had found widespread misconduct by agents working for humana and has ordered the company to take corrective action to protect consumers against high-pressure sales tactics.
the oklahoma state insurance commissioner said that some agents had enrolled medicare recipients in humana products that they did not understand or want.

**PL-FT Summary:**
oklahoma insurance comr kim holland, in first major investigation of medicare marketing, documented widespread misconduct by agents working for humana and has ordered company to take corrective action to protect consumers against high-pressure sales tactics.
says some agents had enrolled medicare recipients in humana products that they did not understand or want.

Figure 3: Two summary examples in NYT test set. The summaries are from the leading three sentences of the article, the reference, the pretrained-only model PL-NoFT and the pretrained+finetuned model PL-FT.

Both PL-NoFT and PL-FT yield more novel 1-grams in summary than the reference. And PL-NoFT has similar novelty ratio with the reference in other n-gram categories. Also, we observe that the novelty ratio drops after finetuning. We attribute this to the strong lead bias in the NYT dataset which affects models trained on it.

## 5.7 HUMAN EVALUATION

We conduct human evaluation of the generated summaries from our models and the pointer generator network with coverage. We randomly sample 100 articles from the CNN/DailyMail test set and ask 3 human labelers from Amazon Mechanical Turk to assess the quality of summaries with a score from 1 to 5 (5 means perfect quality. The labelers need to judge whether the summary can express the salient information from the article in a concise form of fluent language. We put the details of

| Model | Average Score | Standard deviation |
|---|---|---|
| PTGEN+COV | 3.24 | 1.17 |
| PL-NoFT | 3.47 | 1.12 |
| PL-FT | **4.09**** | 0.88 |

Table 5: Average and standard deviations of human evaluation scores for summaries on CNN/DailyMail test set. Scores range from 1 to 5 with 5 being perfect. Each summary is judged by 3 human evaluators. PL-FT's result is statistically significant compared with pointer-generator network with coverage with a p-value less than $10^{-7}$.

evaluation guidelines in Appendix. To reduce bias, we randomly shuffle summaries from different sources for each article.

As shown in Table 6, both of our models PL-NoFT and PL-FT outperform the pointer generator network (PTGen+Cov), and PL-FT's advantage over PTGen+Cov is statistically significant. This shows the effectiveness of both our pretraining and finetuning strategy. To evaluate the inter-annotator agreement, we compute the kappa statistics among the labels and the score is 0.34.

# 6 CONCLUSIONS

In this paper, we propose a simple and effective pretraining method for news summarization. By employing the leading sentences from a news article as its target summary, we turn the problematic lead bias for news summarization in our favor. Based on this strategy, we conduct pretraining for abstractive summarization in a large-scale news corpus. We conduct thorough empirical tests on five benchmark news summarization datasets, including both automatic and human evaluations. Results show that the same pretrained model without any finetuning can achieve state-of-the-art results among unsupervised methods over various news summarization datasets. And finetuning on target domains can further improve the model's performance. We argue that this pretraining method can be applied in more scenarios where structural bias exists.

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

APPENDIX

MODEL SPECIFICATIONS IN FINETUNING

For NYT dataset, we apply a minimum and maximum summary generation length of 50 and 150 sentence pieces, respectively. The article is truncated to the first 400 sentence pieces. The beam width is 8.

For XSum dataset, we apply a minimum and maximum summary generation length of 30 and 150 sentence pieces, respectively. The article is truncated to the first 400 sentence pieces. The beam width is 1.

For CNN/DailyMail dataset, we apply a minimum and maximum summary generation length of 60 and 150 sentence pieces, respectively. The article is truncated to the first 350 sentence pieces. The beam width is 5.

For DUC-2003/2004 datasets, we apply a minimum and maximum summary generation length of 12 and 20 sentence pieces, respectively. The article is truncated to the first 250 sentence pieces. The beam width is 3.

HUMAN EVALUATION SETUP

We ask 3 human labelers from Amazon Mechanical Turk to assess the quality of summaries with a score from 1 to 5. Here's the evaluation guideline shown to the labelers:

| Score | Criteria |
|-------|----------|
| 5 | The summary contains all key points of the news. |
| 4 | The summary misses one key point of the news. |
| 3 | The summary misses two key points of the news. |
| 2 | The summary misses all key points of the news. |
| 1 | The summary is hardly related to the news or the language is not natural and fluent. |

Table 6: Scoring criteria for human evaluation of summaries.

