# OpenReview forum: "Make Lead Bias in Your Favor: A Simple and Effective Method for News Summarization"
_ICLR.cc/2020/Conference — Reject_

### Official Review · AnonReviewer3 · 2019-10-19
**Official Blind Review #3**

**Rating:** 6

**Review:**

This paper proposed an interesting idea on how we can leverage the lead bias in summarization datasets to pretrain abstractive news summarization models on large-scale unlabelled corpus in simple and effective way.

For pre-training, they collected three years of online news articles data. Then, they take the top 3 sentences of the article as summary and the rest of the article as input document. For better choosing such article-summary pairs, they employ effective data cleaning and filtering process. Overall, they collected 21.4M articles for the pretraining.

Overall, the pretrained model does decent on three summarization datasets without any fine-tuning. After fine-tuning the respective datasets, the gains seem significant. Especially on the XSum dataset, the improvements are remarkable.

I believe that the idea is interesting but the experiments are incomplete and more investigation is required to make this paper stronger. Therefore I suggest to reject this paper.

Arguments:
1) The important experiments that are missing in this paper are evaluating the proposed method on better human written summarization datasets -- DUC. The real world summarizations resemble more like the ones in the DUC dataset and it would be interesting to see if the transfer results of the pretrained model on the DUC datasets. The important question is to understand whether the pretrained model which took advantage of lead-bias could achieve good summaries on real summarization samples. This would also answer whether the pretrained just took advantage of the lead-bias issue of many large summarization datasets or does it really learn good summarization model.

2) This paper has good idea but mainly missing ablation studies. For example, how does the proposed model do compared with GPT-2 in the fine-tuning setting, and how do these two models perform on the DUC datasets.

3) During the dataset filtering/collection, a check on the quality of the filtering process by doing a small human study would have been a great addition. Also, instead of showing the output examples (which can go in the supplementary), human study comparing the quality of the pretrained model with fine-tuning and a baseline (can be from previous work) would have been better.

Other minor questions
1) “we only keep articles with 10-150 words in the top three sentences and 150-1200 words in the rest” -- is there any reason on fixing to these numbers? How did you make this decision ?

2) Even though the performance gains look visibly significant, I would suggest to report the statistical significance scores.


**Experience Assessment:**

I have published in this field for several years.

**Review Assessment: Checking Correctness Of Derivations And Theory:**

I assessed the sensibility of the derivations and theory.

**Review Assessment: Checking Correctness Of Experiments:**

I carefully checked the experiments.

**Review Assessment: Thoroughness In Paper Reading:**

I read the paper thoroughly.

---

> ### Author Response · Authors · 2019-11-13
> **Thanks for your comments!**
>
> For your questions:
> 1. We add experiments on DUC-2003 and DUC-2004, using the dataset as test (Table 4). Our pretrained model achieves state-of-the-art results among all unsupervised models.
>
> 2. We add GPT-2's result for CNN/DailyMail and DUC-2003/2004 datasets (Table 3,4). GPT-2 is outperformed by our pretrained-only model PL-NoFT. This is a fair comparison since these two models are both trained only on unlabeled corpus, although GPT-2 has general purposes. Thus, we argue that our pretraining strategy works better in news summarization.
>
> 3. We conduct human evaluation on 100 randomly chosen articles/summaries in CNN/DailyMail dataset and show the results in Section 5.7. Our model outperforms the pointer-generator network and the result is statistically significant.
>
> Answers to minor questions:
> 1. There are a few articles with excessively long content, and we filter them mainly to reduce memory consumption. Also some leading sentences are very short (like "What?") and we filter them as they contain little information and are unlikely to be a good summary. As the pretraining task is very time-consuming, we did not try other settings. We add these information in Section 5.2.
>
> 2. We conduct statistical test on the ROUGE-scores and update all tables. Most of our results are statistically significant with p-value < 0.05, compared with previous best result.

---

### Official Review · AnonReviewer2 · 2019-10-21
**Official Blind Review #2**

**Rating:** 1

**Review:**

This paper suggests generating a large news summarization dataset by taking advantage of the fact that in news articles it is often the case that first few sentences contain the most important information. I have the following criticisms of this paper:
- the idea is not novel. The XSUM dataset cited had used this to create a large dataset based on BBC articles as the editorial guidelines are such that the first sentence is a summary of the article. The lead1 baseline doesn't make sense, as it is the actual reference of the dataset. As implemented, it actually picks the second sentence of the original article, and unsurprisingly works worse than the lead-X for the other two datasets.
- the filtering based on word overlap between the initial sentences and the rest of the document means that the training dataset will encourage models copying words; good summaries don't have high word overlap necessarily.
- no human evaluation is not conducted; ROUGE indicates small differences, but it can't be trusted without confirmation by human evaluation
- I don't agree that using positional information is bad for the models. The point is that we need to do better than that, but we should still take it into account

**Experience Assessment:**

I have published one or two papers in this area.

**Review Assessment: Checking Correctness Of Derivations And Theory:**

N/A

**Review Assessment: Checking Correctness Of Experiments:**

I carefully checked the experiments.

**Review Assessment: Thoroughness In Paper Reading:**

I read the paper at least twice and used my best judgement in assessing the paper.

---

> ### Author Response · Authors · 2019-11-13
> **Thanks for your comments!**
>
> For your questions:
> 1. Our idea is novel in that we use the structural bias in our favor to pretrain a large-scale news summarization model. For XSUM dataset, according to its paper, it uses the accompanying summary which is the first sentence in BOLD font in the article. It is specially editted by editors to summarize an article. So it's a special format of BBC articles which facilitates fast reading, not just the first sentence of the article. Therefore, the LEAD-1 baseline, which is also used in XSUM and BERTSUM papers, is a valid leading part of the article.
>
> 2. The filtering is based on non-stopping words, and the overlapping ratio implies the amount of carried-over information. As an evidence, we compute the overlapping ratio of non-stopping words between golden summary and the article in CNN/DailyMail dataset and the median is 0.87 (which is not surprising due to lead bias). Then we compute the same ratio between the first 3 sentences and the rest of the article in CNN/DailyMail, and the median is 0.77. Thus, a high overlapping ratio is typical for summaries written by human. We add this information in Section 5.2.
>
> 3. We conduct human evaluation on 100 randomly chosen articles/summaries in CNN/DailyMail dataset and show the results in Section 5.7. Our model outperforms the pointer-generator network and the result is statistically significant.
>
> 4. Positional bias helps with fast news reading, and it also eases the creation of news summarization datasets. However, the positional bias lowers the bar for model to comprehend the article for summary generation. Positional bias is not present in many tasks other than news, like document or dialogue transcript summarization. Therefore, we propose our method to take advantage of lead bias and train a model that could summarize based more on the content, instead of the position.

---

### Official Review · AnonReviewer1 · 2019-10-23
**Official Blind Review #1**

**Rating:** 6

**Review:**

For news article it has been know since long that the LEAD baseline is a tough-to-beat competitor. This paper proposes to use this knowledge as self-supervision for training summarization models.
For this the author download and clean 3 years of news articles and use this to (pre-)train a Tranformer model. This alone already provides a competitive baseline, which is greatly improved by fine-tuning it on 3 different data-sets. While the data-set can probably not be released, it would be very helpful to have the model available for reproductivity and benchmarking.

The paper is clear and well-written. Section 4 I believe is very redundant for an ICLR audience and could be moved to the appendix, making space for a more detailed analysis. One criticism is that the paper is light: the author show that a simple idea works (this is a compliment), but I would have expected to have used the remaining space for ablation studies or a discussion on where this leads.
One important point which I would like to see before recommending acceptance is a comparison to know if what is helping is just more data, or the summarization objective. Using lots of more data beats all those numbers (see BERTSUM paper, Liu & Lapata 2019). The comparison I am missing is training BERT on your crawled data-set, and use that for BERTSUM (the code is available). If that helps as much as the summarization pre-training then it would be disappointing but a nice result in favor of language modeling. If not, then it is a strong support for your idea.

Two other points which should at least be discussed, as it gives the impression of cherry-picking results instead:
1/ Table 1 is recall; Table 2&3 F1. Why?
2/ The parameters of fine-tuning of the appendix vary wildly depending on the data-set (in particular, the difference in the width of the beam search is striking). Was this optimized on test-data? What is the sensitivity of the summaries to this?

I do not understand the last two sentences of Sect 4 ("A candidate word leading...). Could you explain?

**Experience Assessment:**

I have published one or two papers in this area.

**Review Assessment: Checking Correctness Of Derivations And Theory:**

N/A

**Review Assessment: Checking Correctness Of Experiments:**

I carefully checked the experiments.

**Review Assessment: Thoroughness In Paper Reading:**

I read the paper thoroughly.

---

> ### Author Response · Authors · 2019-11-13
> **Thanks for your comments!**
>
> For where our result leads, we argue that taking advantage of positional bias or any structural information can help with model pretraining. This has a lot of importance as nowadays large pretrained models have proved to be very effective in NLP. Furthermore, our idea offers a different angle from the masked language model, which is more artificially created.
>
> The idea of pretraining BERT on our large news corpus then using BERTSUM is very good. Due to time limit, we could not finish this experiment before the rebuttal deadline. But we will definitely follow this idea and conduct experiments. In accordance with your idea, we add in results from the GPT-2 model for CNN/DailyMail and DUC2003/2004 (Table 3&4), and it is outperformed by our pretrained-only method PL-NoFT. As both models are pretrained on unlabeled data, this result shows the effectiveness of our approach.
>
> For your questions,
> 1) We use recall on NYT and F1 in XSum/CNN exactly following BERTSUM (https://arxiv.org/pdf/1908.08345.pdf) for fair comparison.
>
> 2) The hyper-parameters are tuned on the validation data using the pre-trained model. Then the finetuned model also takes the same set of hyper-parameters. The result is not very sensitive to these parameters.
> For example, in XSUM, the ROUGE-1 on validation data with different beam-width:
> 1	23.355 (* we choose this)
> 2	22.91
> 3	23.011
> 4	22.391
> 5	22.318
>
> 3) The last but one sentence means trigram blocking: if the next generated word triggers a duplicate trigram, we do not use it. Like "A A B A A", if "B" is the next top word candidate, we ignore it as it will triggle a duplicate trigram AAB.
>    The last sentence means we compute the average cross entropy per word as criterion, which is a popular standard in generation. If the sum is used,
> shorter sentences are favored.

---

> > ### Comment · AnonReviewer1 · 2019-11-14
> > **Human evaluation - weak analysis**
> >
> > re point 1), about Recall vs F1. Can you explain why different scores are used for different corpora? I assume this is due to summaries of different length, but an explanation would be helpful. Just reporting the same measures as previous without critical analysis opens the door to many bad things.
> >
> > The additional experiments on DUC are always good. However, I don't think that too much should be read into the bad performance of GPT-2, as this was not adapted in any way for summarization.
> >
> > The human evaluation is done in a very shallow way, and as it stands does not bring much to make the claims stronger.
> >  - what is the inter-annotator agreement? Without any filtering you risk of running into spammers
> >  - could you report the exact phrasing the annotators saw in the Appendix? Standard practice is to evaluate different aspects separately, as you merged it all together it is not obvious what is being evaluated.
> >
> > I believe this is a valuable paper with a good insight for the summarization community. It is a bit light still, and unfortunately the most interesting question for me (is pre-training for summarization better than pre-training for masked LM, controlling for the data) is not yet answered.

---

> > > ### Author Response · Authors · 2019-11-14
> > > **Thanks for your feedback**
> > >
> > > For your questions:
> > >
> > > 1. Usually ROUGE-F1 is used for summarization. However, on NYT dataset, we align with BERTSUM paper to use limited-ROUGE-recall, which caps the generated summary at the golden summary's length. In this case, ROUGE-recall makes more sense. Similarly, previous papers on DUC2003/2004 cap generated summary at 75 characters (required by DUC competition) and ROUGE-recall has been used for the evaluation in previous literature.
> > >
> > > 2. Yes, GPT-2 is not specifically trained for summarization. But the GPT paper reports their performance on summarization (CNN/DM) so we conduct a comparison against GPT-2.
> > >
> > > 3. For inter-annotator agreement, we add the kappa statistics in Section 5.7, which is 0.34.
> > >    We added in Appendix the evaluation guidelines of scoring criteria we gave to the labelers.
> > >
> > > 4. Due to time constraint, we could not finish the BERT training on our data at this time, since the data we used (21.4M articles) contains more than double the number of tokens used in BERT. But this is a great idea and we will try it in later studies.
> > >    On the other hand, BERTSUM method requires summarization-specific linear layers to be added and trained after masked LM is trained. In comparison, our model is ready to use as soon as the pretraining finished.

---

### Author Response · Authors · 2019-11-13
**Revised paper**

Dear reviewers, we have added the following updates into the revised version of our paper:
1. We add in human evaluation results in Section 5.7.
2. We add experiments on DUC-2003 and DUC-2004 (Table 4).
3. We remove introductory details of transformers from Section 4.
4. We explain our choice of pretraining hyper-parameters in Section 5.2.
5. We add statistical test for all of our results (Table 1-5, **: p-value<0.01, *: p-value<0.05)

---

### Decision · Program_Chairs · 2019-12-19

**Decision:**

Reject

**Comment:**

This paper proposes a method to leverage the Lead (i.e., first sentence of an article) in training a model for abstractive news summarization.

Reviewers' initial recommendations were weak reject to weak accept, pointing out the limitations of the paper including 1) little novelty in modeling, 2) weak evaluation, and 3) lack of deep analysis. After the author rebuttal and revised paper, one of the reviewers increased the score and were leaning toward weak accept.

However, reviewers noted that there was significant overlap with another submission, and we discussed that it would be best to accept one of the two, incorporating the contributions of both papers. Hence, I recommend that this paper not be accepted, and perhaps some of the non-overlapping contents of this paper can be included in the other, accepted paper.

Thank you for submitting this paper. I enjoyed reading it.